# Human Adipose-Derived Stromal Cells Delivered on Decellularized Muscle Improve Muscle Regeneration and Regulate RAGE and P38 MAPK

**DOI:** 10.3390/bioengineering9090426

**Published:** 2022-08-30

**Authors:** Lucas C. Olson, James T. Redden, LaStar Gilliam, Tri M. Nguyen, Josephina A. Vossen, David J. Cohen, Zvi Schwartz, Michael J. McClure

**Affiliations:** 1College of Engineering, Virginia Commonwealth University, Richmond, VA 23284, USA; 2Department of Gerontology, Virginia Commonwealth University, Richmond, VA 23298, USA; 3Department of Orthopaedic Surgery, Virginia Commonwealth University, Richmond, VA 23298, USA; 4Department of Periodontics, University of Texas Health Science Center at San Antonio, San Antonio, TX 78229, USA

**Keywords:** muscle, regeneration, decellularized tissue scaffold, adipose-derived stromal cells

## Abstract

Volumetric muscle loss (VML) is the acute loss of muscle mass due to trauma. Such injuries occur primarily in the extremities and are debilitating, as there is no clinical treatment to restore muscle function. Pro-inflammatory advanced glycation end-products (AGEs) and the soluble receptor for advanced glycation end-products (RAGE) are known to increase in acute trauma patient’s serum and are correlated with increased injury severity. However, it is unclear whether AGEs and RAGE increase in muscle post-trauma. To test this, we used decellularized muscle matrix (DMM), a pro-myogenic, non-immunogenic extracellular matrix biomaterial derived from skeletal muscle. We delivered adipose-derived stromal cells (ASCs) and primary myoblasts to support myogenesis and immunomodulation (N = 8 rats/group). DMM non-seeded and seeded grafts were compared to empty defect and sham controls. Then, 56 days after surgery muscle force was assessed, histology characterized, and protein levels for AGEs, RAGE, p38 MAPK, and myosin heavy chains were measured. Overall, our data showed improved muscle regeneration in ASC-treated injury sites and a regulation of RAGE and p38 MAPK signaling, while myoblast-treated injuries resulted in minor improvements. Taken together, these results suggested that ASCs combined with DMM provides a pro-myogenic microenvironment with immunomodulatory capabilities and indicates further exploration of RAGE signaling in VML.

## 1. Introduction

Skeletal muscle makes up 40% of the body by mass and is a highly regenerative tissue due to its reservoir of muscle satellite stem cells (MuSCs). Extremity trauma, such as the type incurred during warfare, car accidents, tumor resection, and other blunt trauma can result in volumetric muscle loss (VML) injury. In VML injuries, the MuSC pool is depleted through the loss of muscle volume, infringing on muscle’s inherent regenerative capacity [1,2]. In addition, extracellular matrix (ECM) accounts for ~10% of the skeletal muscle mass and coats muscle fibers and MuSCs [3]. Loss of the ECM eliminates a key element in muscle regeneration. Indeed, when the ECM is present following injury, muscle’s regenerative capacity remains intact [4,5]. A strategy to overcome a loss of both the MuSC pool and the ECM is to implant a muscle-derived ECM seeded with muscle-derived stem cells [6]. Our lab has previously established a method to decellularize skeletal muscle, removing the cellular components that would induce an aberrant immune response, and using this as a platform for new cell ingrowth in the injury area [7,8]. Other research groups have also explored the use of decellularized tissues in VML injuries and most have determined that decellularized matrices result in functional fibrosis and not in regeneration [9,10,11], owing a lack luster regeneration to an inability to support MuSC pool expansion. However, our research showed that our DMM preparation is capable of supporting MuSCs and nascent muscle fiber formation within the graft area [8].

Regenerative medicine strategies often use cellular therapies to enhance tissue regeneration, where the stem and progenitor cells used are typically derived from a single source. These cell therapies have focused on myoblasts, satellite cells, mesoangioblasts, pericytes, and mesenchymal stem cells [12,13]. Stem-cell-based therapies represent a promising VML treatment option because they have the ability to self-renew and differentiate into various types of functional progeny, including skeletal myoblasts [14]. Muscle satellite cells are tissue resident stem cells in skeletal muscle and are the primary cell type used to regenerate new muscle fibers. In uninjured muscle, these cells reside underneath the basal lamina in a quiescent state. Following injury, mitogenic signals activate them into a proliferating state, whereby parent satellite cells give rise to daughter cells often referred to as myogenic precursor cells (or myoblasts), and their activation initiates the myogenic program [15]. While these cells are the gold standard for muscle regeneration studies, few have been able to effectively deliver satellite cells to VML injuries without eliminating the inherent ability of satellite cells to proliferate and self-renew, a key advantage in their use.

While satellite cells are crucial for effective muscle regeneration and functional recovery, the number of cells harvested from muscle is far less than other mesenchymal tissues. This may require the addition of other mesenchymal stem cells to regenerate enough tissue for VML. ASCs (Adipose-Derived Stromal Cells) are a potent source of mesenchymal stem cells, but their ability to enhance muscle regeneration has been limited [16,17,18]. ASCs have been shown to support muscle repair [19], and our preliminary data demonstrate expression of myogenic genes, indicating their potential to enter the myogenic program. Despite the work done to characterize an optimal biomaterial delivery system, the current solutions available for VML have had mixed results with low rates of improvement and the risk of adverse immune responses [12]. Thus, the field continues to investigate a variety of potential scaffolds, cells, and molecular signaling solutions.

The challenge with current scaffold options is that some natural polymers are easily degraded, and synthetic materials risk stimulation of a foreign body response [20,21,22]. This study will use DMM which has been shown to provide a suitable cellular environment and remains within the wound site to allow the native tissue to reform in a VML injury [7,8].

Use of ASCs provides an alternative solution with potential to acquire an appropriate number of cells capable of contributing to muscle regeneration. We have data that indicate an ability for ASCs to enter the myogenic program, and these findings were supported by others [12,13,14]. In other recent studies, ASCs cultured with myoblast-conditioned media increased levels of myogenic differentiation [15] or co-cultured with muscle satellite cells stimulated ASC fusion into multinucleated myotubes [16]. In addition, ASCs were implanted into a muscle defect with an angiotensin inhibitor that stimulated ASC fusion into new myofibers [17], suggesting that ASCs can communicate and fuse with muscle progenitors and aid in regeneration of new muscle fibers in VML injuries.

For the current study, we hypothesized that delivering either ASCs or early-stage myoblasts to the injury site using DMM would improve muscle regeneration. In addition to satellite cells, it has been shown that ASCs participate in the myogenic process. Moreover, these cells are easy to harvest from liposuction procedures, making their use in the clinic relevant for skeletal muscle tissue engineering. We used human ASCs or human myoblasts to test this hypothesis in an athymic rat gastrocnemius defect model as a step toward clinical translation in humans. 

## 2. Materials and Methods

*Decellularized muscle preparation.* All animal studies were performed in accordance with approved VCU protocols (IACUC #AD10000675). Male Sprague Dawley rats weighing 250–300 g (Envigo, Huntingdon, UK) were euthanized using CO_2_. Gastrocnemius muscles were isolated bilaterally frozen at −80 °C, and shipped to MTF Biologics (Edison, NJ, USA) to be decellularized. Decellularization was performed via a proprietary method composed of multiple saline, detergent, and disinfection soaks using American Association of Tissue Banking and Food and Drug Administration approved protocols developed by MTF. The rat tissue was processed aseptically and without any terminal sterilization. Frozen decellularized muscle matrices (DMM) were shipped back to Virginia Commonwealth University (VCU) and were kept frozen at −80 °C until surgery.

*Cell culture.* Human myoblasts (Cook Myosite, NJ, USA) were subcultured in myoblast growth media (Cook Myosite, NJ, USA) and passaged at 50% confluence. Cells were maintained in subculture until passage 3 and were used for DMM seeding. Human ASCs were subcultured in αMEM supplemented with 10% fetal bovine serum (FBS), 1% penicillin/streptomycin (P/S), and 1% L-glutamine. Cells were maintained in subculture and passaged at 70% confluence to be seeded onto DMM scaffolds. Cell seeding on DMM was first optimized for cell response data using both murine myoblasts and human ASCs. Murine myoblasts (C2C12, ATCC) were cultured on DMM using DMEM supplemented with 10% FBS and 1% P/S, and hASCs were cultured as described above. An optimal seeding density of 50,000 cells/cm^2^ was determined. Seeding density was checked using Live/Dead Viability staining (Thermo Scientific, L3224). Live/Dead staining was prepared by diluting 20 µL of 2 mM EthD-1 into 10 mL of 1x PBS followed by a dilution of 5 µL of 4 mM calcein AM stock solution into the 4 µM EthD-1 solution. Then, 100 µL of Live/Dead stain was added to each cell seeded sample and incubated at room temperature for 30 min. Labeled cells were imaged using a Zeiss AXIO Observer, Z1 fluorescence microscope. Prior to surgery, DMM was seeded with 2,000,000 cells/scaffold and allowed to subculture for 24 h. At surgery, seeded DMM scaffolds were prepared by washing in 1x PBS just prior to implantation.

*Volumetric muscle loss surgery.* In total, 40 male Foxn1*^RNU^* (RNU) rats (250–300 g, ~4 months old) were obtained from Envigo (Huntingdon, UK) and were divided into five groups (N = 8 rats/group unless specified): sham surgery (n = 6); empty defect surgery, DMM surgery, DMM + hASC surgery, and DMM + h − myoblast surgery. All rats were given ad libitum access to standard pellet and water and provided environmental enrichment. Animals were housed individually. All surgical procedures were performed under an approved protocol at VCU (IACUC #AD10000675) as previously described [8]. Briefly, rats were anesthetized using 4% isoflurane/400 mL/minute O_2_ and prepared for surgery. Rats were transferred to the operating table and anesthesia was continued at 1–3% isoflurane in O_2_. An oblique anterolateral incision extending from the patella to the calcaneus was made. After an incision in the biceps femoris muscle to expose the gastrocnemius was made, the left lateral gastrocnemius muscle was isolated, and a 1.5 × 1 cm defect was cut in the lateral gastrocnemius, taking care to preserve the sural and tibial nerve. At the end of surgery, biceps femoris was sutured closed using 5-0 nylon and skin was stapled closed thereafter.

Sham surgeries were performed as described except when the left lateral gastrocnemius was exposed, no incision was made to create a defect. Instead, biceps femoris and skin were closed without a defect. Empty defect surgeries were performed as described and left untreated, closing the biceps femoris and skin following tissue harvest. Rats that received DMM, DMM + ASC, or DMM + myoblast were sutured into muscle defects, using a modified Kessler technique and taking care to orient the anisotropic features in the direction of the muscle fibers. One rat from the empty defect group and one from the DMM + ASC group died or was euthanized (due to significant weight loss) prior to the end of the study. All remaining rats were euthanized at 8 weeks following muscle physiology tests.

*Ultrasound Imaging.* Rats were anesthetized with isoflurane (4% with 400 mL/min O_2_) and hindlimbs shaved. Ultrasound gel was applied to an L15-7io Broadband compact linear array transducer from Philips (Bothell, WA, USA), and the transducer was applied to the gastrocnemius muscle to visualize the injury area (cross-sectional orientation). Suture landmarks identified the edges of the injury. A cross-section from the injury site’s center was used for analysis. We applied Heckmatt’s Scale [23] to grade the ultrasound images qualitatively. The grading criteria are as follows: Grade I—normal, darker muscle, normal fascia thickness/brightness, and the tibia/fibula will be clearly visible; Grade II—brighter muscle, thicker/brighter fascia, and the tibia/fibula will be clearly visible; Grade III—markedly brighter muscle, much thicker/brighter fascia, and less visible tibia/fibula; Grade IV—very bright muscle, considerably thicker/brighter fascia, and no visible tibia/fibula. Three independent, blinded observers scored each image. The three independent ordinal scores for each image were averaged together, and the resulting data was treated parametrically by applying a one-way ANOVA with Tukey’s post test (α = 0.05).

*Muscle physiology.* Peak tetanic force, peak twitch force, maximum rate of contraction, force–time curve integration, and maximum relaxation were measured. Rats were chosen at random and anesthetized using a vaporizer at 4% isoflurane/400 mL/minute O_2_. Following induction of general anesthesia, the sciatic nerve was isolated, and sural and peroneal branches ligated. Sciatic nerve was then stimulated using platinum electrodes connected to a Grass stimulator model SD9 (Astro-Med, Inc., Westwarwick, RI, USA) at 2 msec duration and 2 msec delay at varying voltages and frequencies. The knee and ankle joints were immobilized, and the Achilles tendon was cut from its insertion and connected to a MLT500/A force transducer (ADInstruments, Inc., Colorado Springs, CO, USA) with 2–0 silk sutures. Output was collected digitally using LabChart 8 software (ADInstruments, Colorado Springs, CO, USA). Optimal muscle length, stimulating voltage, and tetanic frequency were determined. During muscle lengthening, muscle was stimulated to tetanus for 3 s at each interval. Once optimal length was determined, tetanic contraction was stimulated at 3 s intervals until peak tetanic force dropped, indicating fatigue. Immediately following this force drop, 3 separate submaximal stimulations were measured. Those stimulations were used for muscle physiology assessments. Peak tetanic force was measured at maximum force for optimal frequency. Peak twitch force was measured at the peak of the submaximal force–time curve. Measurements in injured and treated limbs were the result of total force output from the posterior crural muscles (medial gastrocnemius, lateral gastrocnemius, soleus, and plantaris).

*Histology.* Whole gastrocnemius muscles were removed and fixed in 10% neutral buffer formalin, dehydrated, and embedded in paraffin. Muscles were cross-sectioned approximately 0.5 cm from the margins. Sections (5 μm) were placed on Histobond slides (VWR, Radnor, PA, USA), deparaffinized and rehydrated, and stained with Masson’s trichrome using Weighert’s hematoxylin (Sigma-Aldrich, St. Louis, MO, USA), Biebrich’s scarlet-acid fuschin (Sigma-Aldrich, St. Louis, MO, USA), and aniline blue (Sigma-Aldrich, St. Louis, MO, USA). Coverslips were mounted with xylene-based mounting media and allowed to dry flat before imaging.

*Histomorphometry.* Histomorphometry was used to quantify the percent of centrally located nuclei, myofiber diameter, and ratio of muscle to collagen. Histological sections stained with Masson’s trichrome were imaged using 10× and 40× objectives, and were assessed as previously reported [8]. Healthy muscle from sham operated animals was used as a positive control. Histological analysis was performed on each muscle (n = 6 for sham, n = 7 for Empty, n = 7 for DMM-ASC, n = 8 for DMM, n = 8 for DMM-Myo) in 3 different locations within the graft area. Field sizes of 650 × 870 μm and 165 × 220 μm were used for histomorphometry. DMM locations were identified by first locating the zone of injury using injury margins. Once the zone was established, images were taken from three locations within the injury site, one at the margin, a second within the middle of the graft, and a third within an additional area of the graft away from the margins.

*Nanostring.* Gene expression of 817 total RNAs were measured using the NanoString fluorescent RNA hybridization and counting method. The custom panel measured genes related to muscle and nerve expression from the mouse genome matching at least 90% homology to the rat genome of interest. Samples were diluted to 55 ng/uL and checked for 260/280 purity using spectrophotometry. In total, 14 genes were used as reference genes while 10 others were used as housekeeping normalization genes. Counting was performed using the nCounter^®^ Digital Analyzer while raw data analysis was accomplished using nSolver^TM^ Analysis Software 4.0. Background thresholding was used to eliminate potential noise caused by low count (<20) genes followed by housekeeping normalization using geometric means. Genes found outside of the normalization factor range of 0.1–10 were flagged for QC. Fold change from each group was calculated against sham. Genes were considered differentially expressed if their log2 fold change was greater than |±2| and *p*-value ≤ 0.05.

*Western blot.* In total, 30 mg of gastrocnemius muscles were homogenized in NP-40 lysis buffer (BP-119, Boston BioProducts, Ashland, MA, USA) with a PI cocktail and 25 mM NaF with a 6.0 mm zirconium bead in a beadbug homogenizer (BeadBug™ Cat #: 31-212, Genesee Scientific, San Diego, CA, USA) at 4000 rpm for 60 s 5 times while keeping the tubes on ice for at least 5 min between runs. The homogenate was centrifuged at 9703 rpm (10,000× *g*) (Centrifuge 5427 R, Eppendorf, Hamburg, Germany) for 10 min, and the supernatant was used for Western blotting. Briefly, the supernatant was run on the BCA assay. Equal amounts of protein were denatured with Laemmli buffer at 100 °C for 10 min, then electrophoresed on polyacrylamide gels, transferred to a PVDF low fluorescence membrane, blocked for 1 h at room temperature, and stained overnight with primary antibodies listed in Table 1. The membranes were then incubated with secondary antibodies (926-68073, 926-32210, and 926-32211, LI-COR Biosciences, Lincoln, NE, USA) for 40 min at room temperature, then they were imaged on the LI-COR Odyssey and quantified. Each blot was analyzed separately and results were normalized to GAPDH levels. The normalization factor for samples on one blot was determined by identifying the highest GAPDH signal and dividing each GAPDH signal intensity by the highest GAPDH signal, producing a value of 1 for the strongest signal and less than 1 relative to that stronger signal. Target protein signals were divided by the normalization factor to obtain the normalized signal.

*AGEs.* Muscle samples were minced and homogenized with the Minute^TM^ Total Protein Extraction Kit for Muscles (using the Denaturing Buffer) and ran on an AGE ELISA (Cell Biolabs, STA-817) to determine AGE levels. For DMM, homogenization was completed with a 6.0 mm zirconium bead in a beadbug homogenizer (Genesee Scientific, BeadBug™ Cat #: 31-212) at 4000 rpm for 60 s 20 times while keeping the tubes on ice for at least 5 min between runs, to pulverize the tough ECM, followed by centrifugation at 13,000 rpm (17,949× *g*) (Eppendorf, Centrifuge 5427 R) for 3 min. The Pierce™ BCA Protein Assay Kit (Thermo Fisher Scientific, 23225 and 23227) and hydroxyproline assays were used for normalization purposes.

*Statistical analysis.* Each variable was tested using N = 8 independent animals. The animal number was chosen based on a power analysis using an alpha of 0.05 and a power of 80% (delta = 5, sigma = 3, m = 1) to reveal a minimum of n = 7 per group to yield statistical significance. Data are presented as mean ± SEM with analysis completed using GraphPad Prism 6.0 (GraphPad, La Jolla, CA, USA). Analyses comparing more than 2 groups used one-way analysis of variance with Tukey post hoc test to determine differences between rat treatments, while comparisons of 2 groups used Student’s unpaired *t*-test. For Heckmatt’s Scoring, the three independent ordinal scores for each image were averaged together, and the resulting data was treated parametrically by applying one-way ANOVA with Tukey’s post hoc test (α = 0.05). All *p* values < 0.05 were considered significant.

## 3. Results

### 3.1. Gene Analysis for VML Injuries Treated with DMM

The effect of DMM in injury sites was compared to injuries with no treatment and sham operated controls. Over 700 genes were assessed using muscle lysates isolated from the VML injury sites. Heat maps demonstrated unique gene profiles between DMM and Empty Defects (Figure 1A). Volcano plots were examined to isolate genes that demonstrated a log 2-fold change (Figure 1B). Evidence showed that Empty Defects had an increased number of fibrotic gene markers compared to DMM (Appendix A). Moreover, one of the primary genes that was isolated during this analysis showed upregulated levels for MAP3K, Jun, and cMyc (Figure 1C). These were mapped using a pathway analysis and indicated increased inflammatory stimulants that were possibly regulated by RAGE and related to p38 MAPK (Figure 1D).

### 3.2. Cell Seeding onto DMM

To assess cellularization of DMM, we seeded murine myoblasts (C2C12s) or hASCs onto DMM and stained the cells using Live/Dead. These first studies demonstrated that myoblasts and hASCs could be successfully grown on DMM. We then determined mRNA levels in myoblasts and hASCs cultured on DMM versus TCPS. mRNA levels in myoblasts showed increased levels for Myod (Figure 2B) and Myog (Figure 2C). All other myogenic markers were unchanged (Figure 2D–F). hASCs cultured on DMM demonstrated elevated levels of MYF5 (Figure 2G) and levels for ITGA7 (Figure 2K) were detectable when compared to TCPS. All other measured genes were not different between DMM and TCPS.

### 3.3. Muscle Force

Muscle force decreased in VML-injured muscles by 23% in empty defect sites, 22% in DMM- and ASC-treated sites, and 12% in myoblast-treated sites when compared to sham animals (Figure 3A). This produced a significant decrease in muscle maximum tetanic force output in the VML-injured muscles compared to sham surgeries. In addition, neither DMM treatment nor cell treatment significantly improved muscle force. Integration of force–time curves as well as the maximum and minimum slopes were calculated with no differences amongst any groups (Figure 3B–D). 

### 3.4. Magnetic Resonance Imaging

Magnetic resonance imaging was used to characterize fibrotic areas over the time course of healing. Heckmatt’s scores demonstrated no change in echogenicity at 2 weeks, increased echogenicity by 4 weeks in myoblast-treated injury sites (Figure 4B), and echogenicity was unchanged after 8 weeks of healing (Figure 4C). Of note, echogenicity was on average elevated in all injury sites compared to sham animals although this was not statistically different.

### 3.5. Histological Assessment

VML-injured sites treated with DMM with or without cells showed areas of regeneration (Figure 5). The number of muscle fibers and Feret fiber diameters were first characterized with no differences amongst the groups (Figure 5F,G). Feret’s fiber diameter was plotted as a histogram, showing that sham animals had larger muscle fibers than injured animals with a rightward shift in the distribution (Figure 5H,I). As previously shown in prior research [8], DMM supported de novo fiber formation within the graft area with centrally located nuclei within the graft area. Empty defect sites also showed areas of centrally located nuclei that were similar in number to DMM, but it should be noted that these areas were located at the margins of the injury with no grafted area. In contrast, DMM-treated sites showed newly regenerated fibers within the graft area, away from the margins. ASCs delivered to VML injuries improved the number of regenerated fibers compared to empty injury sites but were not different from DMM or myoblast delivery (Figure 5J). Collagen was assessed using the aniline blue stain and showed increased collagen within all injured muscles compared to sham animals (Figure 5K). 

### 3.6. Myofibrillar Assessment

Human myoblasts and ASCs behaved differently when delivered to muscle injuries. Myoblasts delivered to VML injuries did not affect myosin heavy chain levels when compared to either empty defects or sham animals. In contrast, ASCs delivered to VML injuries increased levels of mixed myosins, which included both slow and fast twitch MyHC (Figure 6A). Those myosins were parsed using more specific antibodies for fast, neonatal, and slow myosin, demonstrating a significant increase in fast myosin that was over 4-fold higher that sham and over 7-fold higher than empty defect surgeries (Figure 6B). Neonatal myosins were unchanged regardless of treatment, but levels in ASC-treated sites were noteworthy with a 3-fold increase over sham and a 9-fold increase over empty defects (Figure 6C). Lastly, slow myosin heavy chains were unchanged regardless of treatment (Figure 6D). 

### 3.7. AGEs and RAGE

Previously, we showed that AGE crosslinks were associated with age-related fibrosis 9 [24]. Whether this was also the case in VML injuries had not yet been determined. Collagen levels were assessed using hydroxyproline assays. Whilst no significant differences in collagen were detected amongst all groups, average levels were more consistently elevated in DMM-, ASC-, and myoblast-treated injury sites (Figure 7A). Interestingly, AGEs were normalized to hydroxyproline and showed a reduction in all treated injury sites using DMM with or without cells (Figure 7B). RAGE levels showed an increase in ASC-treated injuries compared to empty, DMM, and myoblast groups (Figure 7C), and p38 MAPK was most elevated in the DMM-treated VML injuries while levels were reduced in ASC and myoblast groups compared to DMM (Figure 7D).

## 4. Discussion

In this study, we first assessed differences in gene expression amongst DMM-treated VML sites, ED VML sites, and sham operated animals and determined that MAPK signaling was strongly affected including inflammatory associated pathways. These initial studies helped to identify specific pathways related to our injury model. DMM scaffolds were then seeded with human ASCs or myoblasts to determine if delivery of stem cells or progenitor cells would improve muscle regeneration in a VML model. While functional differences were unremarkable, differences in centrally located nuclei were elevated in ASC-delivered cells versus DMM and empty defect animals. In contrast, myoblast-delivered injury sites were unchanged compared to DMM and lower than empty defect animals. Western blotting confirmed the observed increase in central nuclei in ASC-treated injuries, showing increased fast twitch myosin heavy chain. In addition, collagen levels were quantified using Masson’s Trichrome staining and showed increased collagen across all injury sites. We hypothesized that increased fibrosis was related to aberrant AGE/RAGE signaling based on our initial gene analysis of VML wounds treated with DMM. Interestingly, AGE cross-links were downregulated in injured animals while RAGE signaling increased in ASC-treated sites. These data correlated with increased regenerative markers in ASC-treated injury sites, suggesting that RAGE signaling occurred in a normal physiologic manner. Downstream p38MAPK signaling was assessed and found to be lower in DMM- and myoblast-treated animals but similar to empty and sham in ASC, further confirming normal RAGE signaling. This is significant because other studies indicated that muscle trauma aberrantly increases RAGE signaling [25,26], contributing to a prolonged immune response; however, our data show a different effect in a rodent VML model. Overall, our data suggest that ASCs improved levels of muscle regenerative markers compared to myoblasts and that RAGE signaling also appears to be involved in ASC-mediated regeneration.

ASCs are an abundant source of multipotent mesenchymal stromal cells with myogenic potential. These cells are capable of expressing several myogenic factors including Pax7, Myf5, MyoD, and myogenin. Indeed, Di Rocco et al. showed that a small population of ASCs was capable of sporadically converting into a myogenic lineage, suggesting an inherent myogenic potential [14]. Moreover, this same group demonstrated that ASCs injected into an ischemic injury model fused with existing muscle fibers, supporting the idea that ASCs are a suitable stem cell source for muscle regeneration studies. Prior studies also compared ASCs and myoblasts to each other to determine any differences in their regenerative potential when seeded onto a decellularized bladder matrix [27]. While the study did not remark about ASCs versus myoblasts, they demonstrated histological evidence that ASCs and myoblasts were identified in the decellularized matrix.

We first demonstrated that ASCs and myoblasts could be cultured on DMM, and that ASCs expressed myogenic markers when seeded on DMM in vitro. We then determined whether ASCs or myoblasts would change the muscle’s response to VML injury and showed that regenerative markers were unaffected by myoblast seeding whereas ASCs showed improvements in regeneration when assessing MyHC II, centrally located nuclei, and RAGE signaling.

Myosin heavy chains follow a re-expression pattern during muscle regeneration that is important when analyzing de novo muscle fiber growth in a scaffold area [28]. In cardiotoxin-induced injury models, neonatal MyHC was detected within 2–3 days after injury and persisted for up to 3 weeks [29]. The switch from neonatal to adult fast myosins is independent of innervation [30] and is a likely cause for the increase observed in ASC-treated sites. Since cardiotoxin injuries are a fully regenerative model, it is not surprising for us to detect neonatal MyHC in injured critical-sized muscle injury 8 weeks after implantation. Moreover, only in the presence of nerve is slow myosin upregulated and fast downregulated, suggesting that while we showed elevated levels of neonatal and fast myosins in ASC-treated sites they likely remained denervated.

We explored the role of advanced glycation end-products in our VML injury model and its involvement in fibrosis. Previously, we showed that AGEs were associated with fibrosis in older muscle [24]. Those data coupled with our pathway analysis (Figure 1) led us to hypothesize that fibrosis development in VML was also associated with AGEs. Other research showed that AGEs were elevated following chronic muscle injury with long-term RAGE signaling triggered by high concentrations of RAGE ligands producing a deleterious effect [26]. We first tested AGE cross-link levels and determined that AGEs per collagen were suppressed in all VML groups. Moreover, AGEs levels in whole muscle lysates were lower in DMM-treated sites but were at sham levels for all other groups. This suggested that RAGE ligands were not chronically elevated as shown in prior literature studies, where RAGE ligands were explored in myopathies [31,32].

Interestingly, RAGE was elevated in ASC-treated injury sites compared to all other groups. These data coupled with low AGEs and elevated levels of regenerative markers suggested normal physiologic RAGE signaling in the newly regenerating areas. This is supported by studies that demonstrated delayed regeneration in an acute injury model using Ager knockout mice compared to other studies that explored chronic pathologic conditions [26]. These opposing effects were ascribed to the levels of RAGE ligands available to activate the receptor. In addition, RAGE is highly dependent on its co-activators that help direct a particular signaling pathway. We explored whether p38 MAPK was involved given its known role in AGE/RAGE/p38 MAPK signaling [33], and determined that p38 MAPK was elevated in DMM- and ASC-treated injury sites compared to empty defect while myoblast-treated sites were similar to empty.

P38 MAPK plays a critical role in muscle regeneration [34]. It was first described as a transducer of the response to environmental stress conditions, and in muscle was found to regulate muscle fiber formation via satellite cell differentiation, slow myosin heavy chain gene repression [35], and is involved in muscle pathologies [36]. When taken into context with our observed increase in fast MyHC, centrally located nuclei, and RAGE signaling, use of p38 MAPK as a marker of regeneration in a VML model becomes more intriguing. As a disease mechanism, cell delivery reduced p38 levels and this could aid in regeneration [37]. As a repressor of slow myosin heavy chain gene expression, p38 MAPK could explain the increases observed in fast myosin heavy chain protein levels in DMM-, ASC-, and Myo-treated injuries compared to empty sites, but more study would be needed to fully elucidate the role of p38 MAPK in a VML model.

## 5. Conclusions

Prior research has shown that cell delivery to a VML injury site is essential to assist in muscle regeneration, improving the number of de novo muscle fibers within the injury. Our results demonstrated that our DMM was sufficient to improve muscle fiber regeneration and DMM was enhanced by ASCs but not myoblasts. We also determined that AGE/RAGE is not involved in late-stage healing and fibrosis following VML injury, indicating that AGE crosslinks do not adversely affect skeletal muscle ECM. p38 MAPK plays a critical role in muscle regeneration, and its levels were higher in DMM-treated muscles without cells compared to untreated and cell-treated injury sites. We theorize that this response is due to an enhanced myogenic capacity of ASC seeded DMM, and RAGE and p38 MAPK appeared to be regulated independent of one another. In addition, RAGE appeared to function in a pro-myogenic fashion, which was reported to occur under normal physiologic conditions. This could mean that DMM mitigates pro-inflammatory responses and promotes a pro-myogenic environment.

## Figures and Tables

**Figure 1 bioengineering-09-00426-f001:**
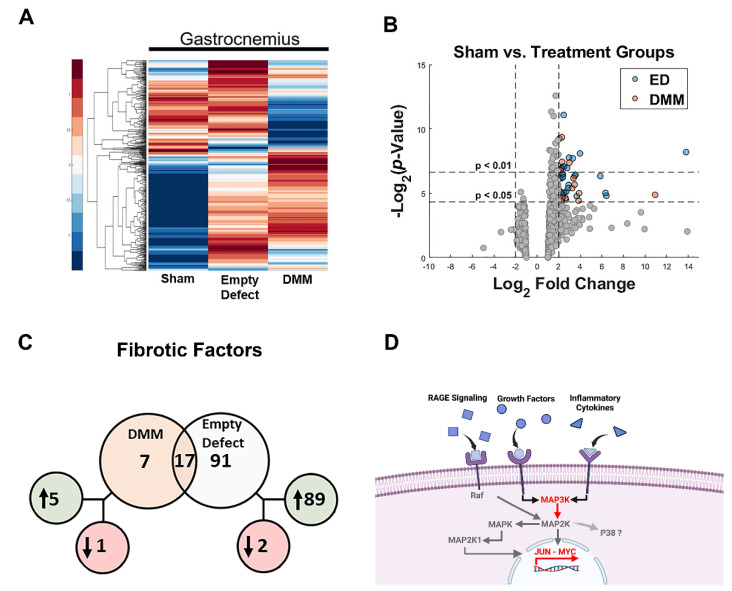
Gene expression analysis of VML injury sites. Nanostring analysis demonstrated unique changes between Sham vs. Empty Defect and Sham vs. DMM (**A**). Gene targets were identified using volcano plots (**B**) and Venn diagrams (**C**). Venn diagram demonstrated shared genes (overlapping circles) versus unshared genes (separate circles). Genes in green were upregulated while genes in red were downregulated. These genes of interest in addition to pathway analysis suggested increased inflammatory pathways which possibly included RAGE (**D**).

**Figure 2 bioengineering-09-00426-f002:**
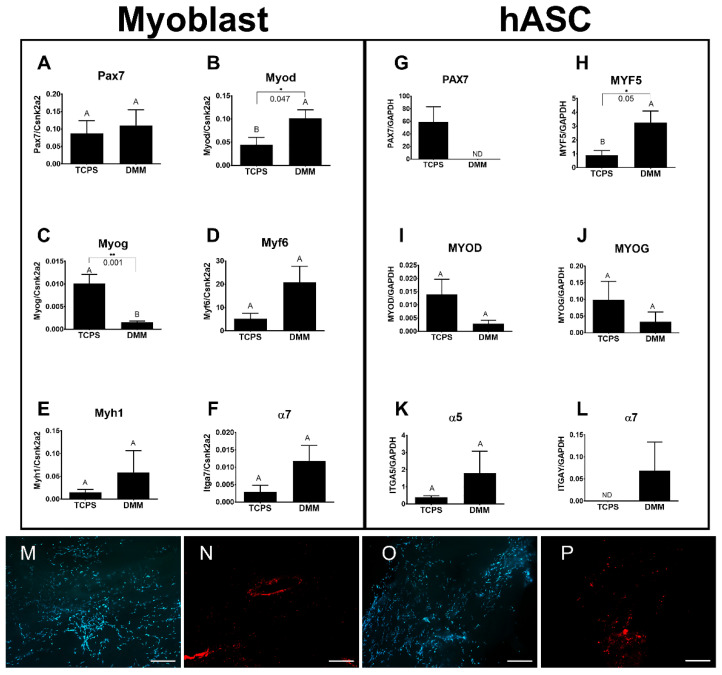
Gene expression analysis and Live/Dead staining for myoblasts and ASCs cultured on DMM. Myoblasts and ASCs were seeded onto DMM and assessed for markers of differentiation. No change in Pax7 was detected (**A**) while Myod (**B**) and Myog (**C**) increased compared to TCPS. Late markers for Myf6 (**D**), Myh1 (**E**), and α7 (**F**) were not different from TCPS. ASCs cultured on DMM did not express PAX7 (**G**) but did express higher levels of MYF5 (**H**). No change in MYOD (**I**), MYOG (**J**), or α5 (**K**) were detected. α7 (**L**) was detected in ASCs cultured on DMM while it was not detected on TCPS. Myoblasts and hASCs were seeded onto DMM and stained for Live/Dead (**M**–**P**). Letters not shared indicate a significant difference (*p* < 0.05, unpaired *t*-test). Expression not detected was marked as “ND”. Data shown are means ± SEM of 6 samples from a representative experiment. Each DMM was randomly selected from a different donor. Experiments were repeated to ensure validity. Scale bar size is 500 µm.

**Figure 3 bioengineering-09-00426-f003:**
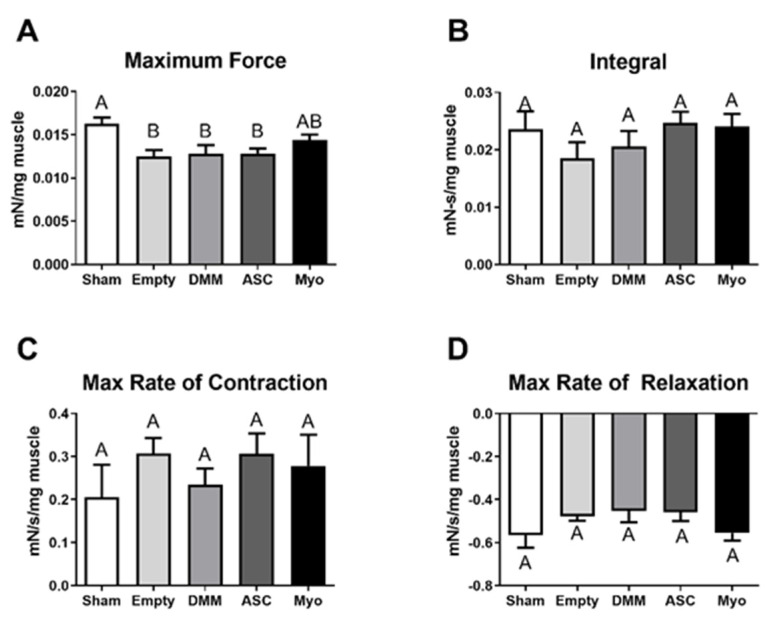
Muscle force analysis demonstrates no overall improvements after cell treatment. Maximum muscle force output from the posterior crural muscles was analyzed and showed reduced force in injured limbs but no improvements were detected in treated injury sites (**A**). Force–time integration (**B**), max rate of contraction (**C**), and max rate of relaxation (**D**) also showed no differences amongst treatment groups and no groups were different from sham surgeries. Letters not shared indicate a significant difference (*p* < 0.05, Tukey). Data shown are means ± SEM of 8 animals.

**Figure 4 bioengineering-09-00426-f004:**
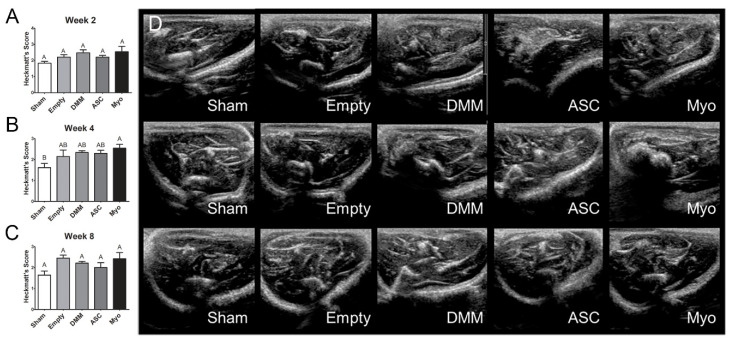
Echogenicity assessment using a Heckmatt’s score. Muscles were evaluated for echogenicity and scored at weeks 2 (**A**), 4 (**B**), and 8 (**C**). Representative images of each group are shown in panel (**D**). Letters not shared indicate a significant difference (*p* < 0.05, Tukey). Data shown are means ± SEM of 8 animals.

**Figure 5 bioengineering-09-00426-f005:**
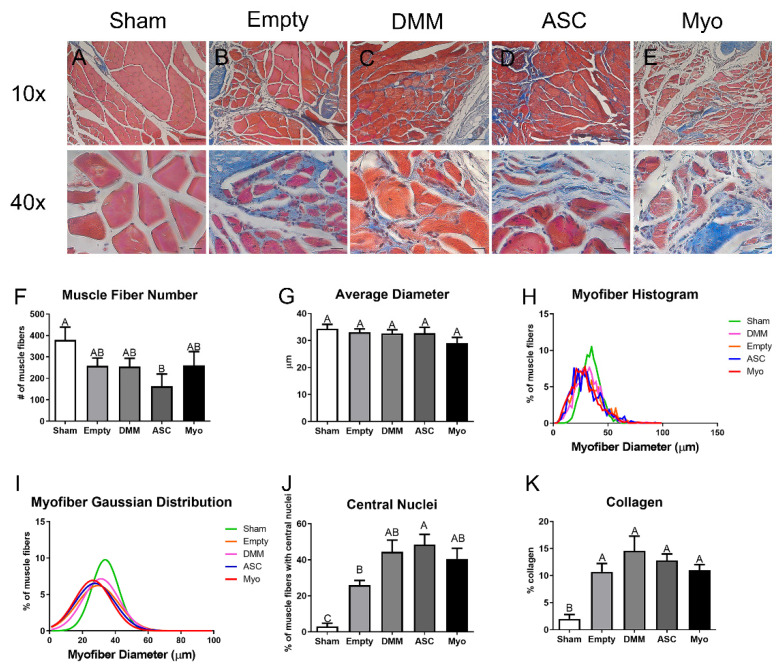
Histological staining and morphometric analysis for VML injuries treated with myoblasts and ASCs. Representative images demonstrate a normal appearance of muscle in sham operated animals (**A**), while a response to injury and treatment is represented in (**B**–**E**). Histomorphometry showed a similar number of muscle fibers (**F**) and a similar Feret’s diameter (**G**). Histograms reveal a slight leftward shift in Feret’s diameter (**H**,**I**). Increased central nuclei were detected in all groups compared to sham while ASC-treated injuries showed the largest increase (**J**). Lastly, increased collagen levels were determined relative to sham for all injury sites (**K**). Letters not shared indicate a significant difference (*p* < 0.05, Tukey). Data shown are means ± SEM of 8 animals. Scale bars for 10× and 40× images are 100 and 20 µm, respectively.

**Figure 6 bioengineering-09-00426-f006:**
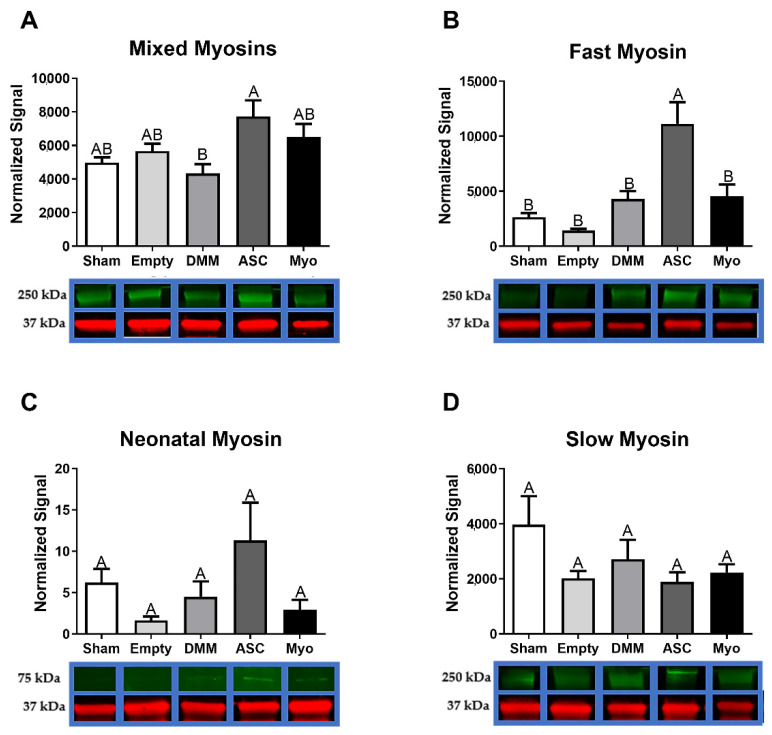
ASC-treated DMM support increased fast myosin levels. Myosin heavy chains were analyzed for a mixed population of slow and fast myosins (**A**), fast myosin heavy chain (**B**), neonatal myosin (**C**), and slow myosin heavy chain (**D**). Protein levels for mixed slow and fast twitch myosin heavy chain were elevated in ASC-treated injuries, and DMM-treated animals were lowest. Fast twitch MyHC showed increased levels only in ASC-treated animals. Neonatal and slow myosin levels were unaffected. Data shown are means ± SEM of 8 animals. Western blot bands are representative bands where green represents the target antibody and red represents GAPDH for normalization. Western blots were repeated to ensure validity. Letters not shared indicated a significant difference (*p* < 0.05, Tukey).

**Figure 7 bioengineering-09-00426-f007:**
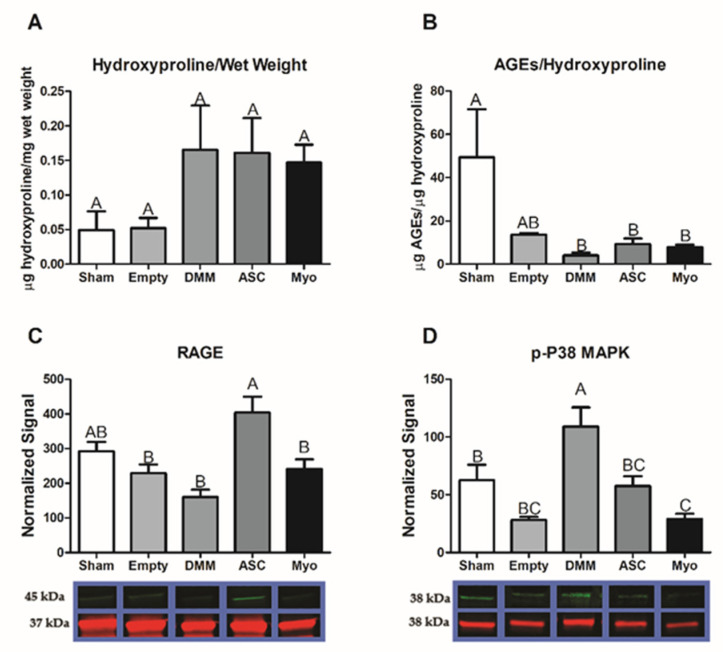
AGEs do not mediate fibrosis and RAGE is activated normally in VML injuries. Collagen levels measured by hydroxyproline were unchanged across all groups (**A**). AGEs were normalized to hydroxyproline and showed reduced levels in DMM-, ASC-, and myoblast-treated groups (**B**). RAGE levels were higher in ASC-treated injuries, but similar to sham (**C**). p38 MAPK was elevated in DMM-treated sites (**D**). Data shown are means ± SEM of 8 animals. Hydroxyproline, AGE ELISA, and Western blots were repeated to ensure validity. Western blot bands are representative bands where green represents the target antibody and red represents GAPDH for normalization. Letters not shared indicated a significant difference (*p* < 0.05, Tukey).

**Table 1 bioengineering-09-00426-t001:** List of antibodies used for Western blotting.

	Host Species	Antibody, Clone	Cat. No.	Company
1	Mouse	αMyHC, A4.1025	05-716	Sigma Aldrich, St. Louis, MO, USA
2	Mouse	αMyHC Fast, MY-32	M4276	DHSB, Iowa City, IA, USA
3	Mouse	αMyHC-I, BA-D5	BA-D5	DHSB, Iowa City, IA, USA
4	Rabbit	αRAGE	ab37647	Abcam, Cambridge, UK
5	Rabbit	αP-38 MAPK	9212S	Cell Signaling, Danvers, MA, USA
6	Mouse	αpP-38 MAPK (Thr180/Tyr182), 28B10	9216S	Cell Signaling, Danvers, MA, USA
7	Rabbit	αGAPDH, 14C10	2118S	Cell Signaling, Danvers, MA, USA

## Data Availability

Data presented in this study are openly available in FigShare, https://doi.org/10.6084/m9.figshare.20566797.v1, https://doi.org/10.6084/m9.figshare.20566602.v1, https://doi.org/10.6084/m9.figshare.20566683.v1, https://doi.org/10.6084/m9.figshare.20566707.v1, https://doi.org/10.6084/m9.figshare.20566731.v1, https://doi.org/10.6084/m9.figshare.20566749.v1, https://doi.org/10.6084/m9.figshare.20566779.v1, accessed on 16 August 2022.

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
