# Peer review of "Human Adipose-Derived Stromal Cells Delivered on Decellularized Muscle Improve Muscle Regeneration and Regulate RAGE and P38 MAPK"

_bioengineering, 2022, doi:10.3390/bioengineering9090426_

Round 1
Reviewer 1 Report (Previous Reviewer 3)
The manuscript has significantly been improved. Lacking descriptions of methods are added now. I have only minor concerns:
line 21: "(N=8rats/group)" insert blank
line 103: "...FBS...P/S" explain abbreviations
line 107: "Seeding 107 density was checked using Live/Dead Viability staining (Thermo Scientific)" short description of the staining procedure?
Figure 2 M-P: add scale bars.
Legend of Figure 5 and 6 should be directly below the figure. I could not find sizes of the scale bars in Figure 5.
Author Response
line 21: "(N=8rats/group)" insert blank
Line 21 was edited to be "(N=8 rats/group)"
line 103: "...FBS...P/S" explain abbreviations
Definition of abbreviations are now included.
line 107: "Seeding 107 density was checked using Live/Dead Viability staining (Thermo Scientific)" short description of the staining procedure?
Authors edited the section on Live/Dead staining to include concentrations of EthD-1 and calcein AM and incubation time for cells.
Figure 2 M-P: add scale bars.
Scale bars are now included along with a description of scale size in the figure legend.
Legend of Figure 5 and 6 should be directly below the figure. I could not find sizes of the scale bars in Figure 5.
A description of scale bar size is now included in the figure legend.
Reviewer 2 Report (Previous Reviewer 4)
Authors have revised the manuscript quite nicely and incorporated suggested changes in the manuscript. This manuscript maybe accepted now for the publication.
Author Response
The authors would like to thank the reviewer for their time, effort, and thorough review.
Reviewer 3 Report (Previous Reviewer 1)
The authors have modified the manuscript as per the reviewer's comment. The manuscript can be accepted for publication.
Author Response
The authors would like to thank the reviewer for their time, effort, and thorough review.
This manuscript is a resubmission of an earlier submission. The following is a list of the peer review reports and author responses from that submission.
Round 1
Reviewer 1 Report
The authors have nicely performed the experiment to demonstrate the utilization of ASCs incorporated DMM for VML defect regeneration and established that DMM has an immense utility to improve muscle fiber regeneration.
However, there have some concerns that the authors need to address before the manuscript can proceed.
1. The authors have performed RNA-seq analysis with Heat maps, volcano plots as well Nanostring analysis. However, the method of this experiment is absent in the materials and method section of the manuscript.
2. In Fig. 2, authors have performed Live-Dead assay, but no explanation in the figure caption (Fig. 2 M & N). In Fig. 2G, authors should define the PAX7 expression in DMM.
3. The authors have performed alcian blue staining to assess the collagen distribution in DMM, ASC, and Myo. However, almost similar alcian blue expressions have depicted in the empty sample as compared to ASC and Myo samples. The authors need to check the result.
4. In the materials & Method section, the authors have mentioned the H&E staining and Masson’s Trichrome (MT) staining for quantifying the collagen level. In the result section (Figure 5), only Alcian blue staining data has been included. However, the author should include the H&E and MT staining data in the manuscript.
5. Authors have performed Western Blot analysis to confirm the central nuclei in ASC treated injuries. However, the author should include the WB band in the manuscript.
All the experiments have been performed for 8 weeks of study, but the authors have included 20-month samples to run the WB gel (Line: 198). Need to clarify the necessity of 20 moths sample.
6. Spelling mistakes has found in 1st line of the abstract, ‘mjuscle’, line: 321 ‘hyroxyproline’, line: 112, incomplete sentence.
7. Authors have performed statistical analysis with ANOVA. However, statistical significance is not clearly represented in the graphs. Therefore, the authors need to mention the exact p-value in the graph.
Author Response
- The authors have performed RNA-seq analysis with Heat maps, volcano plots as well Nanostring analysis. However, the method of this experiment is absent in the materials and method section of the manuscript.
We added a section on Nanostring analysis to the methods.
- In Fig. 2, authors have performed Live-Dead assay, but no explanation in the figure caption (Fig. 2 M & N). In Fig. 2G, authors should define the PAX7 expression in DMM.
Authors added an explanation of Live Dead staining and defined N.D. as not detected in the caption as well.
- The authors have performed alcian blue staining to assess the collagen distribution in DMM, ASC, and Myo. However, almost similar alcian blue expressions have depicted in the empty sample as compared to ASC and Myo samples. The authors need to check the result.
We agree that all injury sites had similar blue staining and this was acknowledged in the results and discussion section. To help clarify this, we included that injured sites were compared to sham animals. In addition, we corrected an error with the staining - alcian was changed to aniline.
- In the materials & Method section, the authors have mentioned the H&E staining and Masson’s Trichrome (MT) staining for quantifying the collagen level. In the result section (Figure 5), only Alcian blue staining data has been included. However, the author should include the H&E and MT staining data in the manuscript.
Authors removed the statement that H&E tissue sections were stained as this was not done originally.
- Authors have performed Western Blot analysis to confirm the central nuclei in ASC treated injuries. However, the author should include the WB band in the manuscript.
Western blot bands were included in the manuscript.
All the experiments have been performed for 8 weeks of study, but the authors have included 20-month samples to run the WB gel (Line: 198). Need to clarify the necessity of 20 months sample.
The reviewer is correct that any aging study was not a part of this study. We removed this line from the manuscript.
- Spelling mistakes has found in 1stline of the abstract, ‘mjuscle’, line: 321 ‘hyroxyproline’, line: 112, incomplete sentence.
Authors went through the manuscript, checked, and correct all spelling errors.
- Authors have performed statistical analysis with ANOVA. However, statistical significance is not clearly represented in the graphs. Therefore, the authors need to mention the exact p-value in the graph.
P values less than 0.05 were considered significant, and this was identified in the materials and methods section. In addition, each figure legend has a description that states, “Letters not shared indicate a significant difference (p<0.05, Tukey).” Using this lettered scheme is an appropriate way to describe statistical significance without the use of specific p-values for each graph. To help further clarify which genes in Figure 2 were significant we included a line and p-values. However, including lines and p-values in bar charts that include all 5 groups is dense and hard to read.
Reviewer 2 Report
The authors tried to present DMM as a scaffold to restore muscle function. However, the story gets diluted with no significance in the muscle strength or collagen. There is a mismatch with the title and the results and conclusion. I have the following queries for the authors.
1. There are too many spelling errors in the abstract, authors need to correct these errors
2. It would be nice if the introduction is re-written, the current form is not organized very well
3. Line 79-81 where the authors say that DMM has been shown to provide a suitable cellular … lacks a reference
4. Line 112 has an empty space Human ASCs were subcultured in _____
5. MAPK signalling is associated with several other signalling, how did you arrive at RAGE signalling other than using pathway analysis did you find related upstream genes in the gene analysis results?
6. Figure 2 what live dead staining method was followed, Assuming two different things we need to see two different images of live and dead cells, the figure 2 has too much background, can you get a clear pic and also present live dead staining with 2 different staining?
7. Does fibrosis not increase muscle force? If so why there was not much increase in the muscle force in any of the conditions?
8. How did you measure the muscle force on DMM?
9. There is no difference in the number of regenerated fibers between the DMM and VML injury but there is increase in central nuclei what is the correlation between these two observations?
10. The authors have not show any of the blot images, Why?
Author Response
- There are too many spelling errors in the abstract, authors need to correct these errors
All spelling errors were corrected.
- It would be nice if the introduction is re-written, the current form is not organized very well
We reorganized and edited the introduction.
- Line 79-81 where the authors say that DMM has been shown to provide a suitable cellular … lacks a reference
Reference was added.
- Line 112 has an empty space Human ASCs were subcultured in _____
The empty space was deleted and alpha MEM included.
- MAPK signalling is associated with several other signalling, how did you arrive at RAGE signalling other than using pathway analysis did you find related upstream genes in the gene analysis results?
Indeed, MAPK signaling is associated with several other signaling pathways. Our pathway analysis determined downstream targets, and those targets were linked to three major membrane associated pathways growth factor-RTK, interleukins, and RAGE. Based on our prior findings and its lack of exploration in VML, we chose RAGE as our target. We have now clarified this in the manuscript.
- Figure 2 what live dead staining method was followed, Assuming two different things we need to see two different images of live and dead cells, the figure 2 has too much background, can you get a clear pic and also present live dead staining with 2 different staining?
Figure 2 was edited to include both live and dead images. Background on the images was reduced as well.
- Does fibrosis not increase muscle force? If so why there was not much increase in the muscle force in any of the conditions?
This is an excellent point. Indeed, fibrosis does increase muscle force; however, in this case the limited increase in muscle force observed in all groups compared to empty defect is likely an inherent limitation of the gastrocnemius model itself and the way muscle force is measured. This is because the collective muscle force from the posterior crural muscles includes the soleus, plantaris, and both the medial and lateral gastrocnemius heads. The only injured site is in the lateral gastrocnemius, allowing for compensation to occur in all other muscle groups. Future work will need to investigate more accurate methods to measure muscle force by severing the medial gastrocnemius, soleus, and/or plantaris tendons. We edited the manuscript to help clarify which muscles were responsible for force output.
- How did you measure the muscle force on DMM?
The section on “Muscle Physiology” was edited in the Materials and Methods section.
- There is no difference in the number of regenerated fibers between the DMM and VML injury but there is increase in central nuclei what is the correlation between these two observations?
These observations were described in the results section. “As previously shown in prior research7, DMM supported de novo fiber formation within the graft area with centrally located nuclei within the graft area. Empty defect sites also showed areas of centrally located nuclei that were similar in number to DMM, but it should be noted these areas were located at the margins of the injury with no grafted area. In contrast, DMM treated sites showed newly regenerated fibers within the graft area, away from the margins.” It is our opinion that this accurately describes the limitation of assessing Empty Defect sites and comparing those to DMM treated sites.
- The authors have not show any of the blot images, Why?
Blot images are now included in the manuscript.
Reviewer 3 Report
The manuscript describes an interesting study, however, the method section is incomplete. I could not find methodology of MRI, life death images, RNA/gene expression, bioinformatics. Was the survival of human cells in the rats muscle defect checked? Was the seeding efficacy checked? the labeling of many figures is too small (e.g. Fig. 2)
The usage of the decellularized ECM should be mentioned in the title.
line 34 "a critical piece" better to describe the mean percentage of ECM in muscle
line 16: "increate" rather "increase"
line 11: "mjuscle" correct: muscle
line 34 "a critical piece" better to describe the mean percentage of ECM in muscle
line 42: "results" rather "result"?, why result decell matrices rather in "functional fibrosis". What means
line 101: provide approximate age of rats.
line 112: provide lacking text segment.
RNU (line 118) explain abbreviation
"prepped" line 125: what does it mean?
line 138: reason of death?
line 171: except for... in Fig. 5 is histomorphology shown for all groups
list all the antibodies in a table.
Figure 1: were are the methods described?
line 230: murine myoblast, method section describes only human-derived
Fig. 2: samples derived from 6 donors?
explain all abbreviations
line 264: Heckmatts score: reference required
line 287 this is not alcian blue stain!
Fig. 5 labeling too small
legend: please correct opereated
line 339 and 342: centrally located nuclei: is redundant
legend of Fig. 7: provide method in the legend and in the method sectuion
line 385: where is the method described.
Author Response
The manuscript describes an interesting study, however, the method section is incomplete. I could not find methodology of MRI, life death images, RNA/gene expression, bioinformatics. Was the survival of human cells in the rats muscle defect checked? Was the seeding efficacy checked? the labeling of many figures is too small (e.g. Fig. 2)
The Materials and Methods section was edited to describe methods used to perform ultrasound, live dead staining, Nanostring analysis, and bioinformatics. Font size of labels in Figure 2 were increased. Unfortunately, the authors did not check the survival of the human cells within the defect area.
The usage of the decellularized ECM should be mentioned in the title.
We edited the title to include decellularized muscle ECM.
line 34 "a critical piece" better to describe the mean percentage of ECM in muscle
We edited this for clarity.
line 16: "increate" rather "increase"
This spelling error was corrected.
line 11: "mjuscle" correct: muscle
This spelling error was corrected.
line 42: "results" rather "result"?, why result decell matrices rather in "functional fibrosis". What means
Functional fibrosis was defined in prior manuscript published by Benjamin Corona who determined that increases in force in VML models were primarily due to the development of “functional fibrosis” and not newly regenerated muscle fibers. Reference to this manuscript was provided in the original submission.
line 101: provide approximate age of rats.
Rat ages are now included.
line 112: provide lacking text segment.
Text is now included.
RNU (line 118) explain abbreviation
Animal model description is now included.
"prepped" line 125: what does it mean?
Prepped changed to prepared.
line 138: reason of death?
Significant weight loss was the reason animals were euthanized or died prior to being euthanized in this study. This is now included in the manuscript.
line 171: except for... in Fig. 5 is histomorphology shown for all groups
All n values were edited in this section.
list all the antibodies in a table.
All antibodies are now listed in Table 1.
Figure 1: were are the methods described?
Methods are now described.
line 230: murine myoblast, method section describes only human-derived
Murine myoblast culture conditions are now described in the methods.
Fig. 2: samples derived from 6 donors?
This is clarified in the figure legend.
explain all abbreviations
line 264: Heckmatts score: reference required
Reference is now included.
line 287 this is not alcian blue stain!
Alcian blue was changed to Aniline blue.
Fig. 5 labeling too small
Font size of Figure 5 was increased.
legend: please correct opereated
This was corrected.
line 339 and 342: centrally located nuclei: is redundant
This was edited for clarity.
legend of Fig. 7: provide method in the legend and in the method section
Methods for figure 7 are now described within the methods section.
line 385: where is the method described.
Methods are now described.
Reviewer 4 Report
This is very interesting manuscript where authors reported that ASCs are known to prevent AGE formation. Study showed an improved muscle regeneration in ASC treated injury sites and signalling pathways. Overall, the manuscript is written well, however I have some minor comments-
1. Abstract should be improved and more structured by considering background, aims & objectives, methodology, results and conclusion. Background is written too long, which is not needed. Add few lines more about results.
2. Authors have presented their study quite well. Even though, i believe a graphical abstract or a figure showing mechanistic view of the present in the main text would be useful to add.
3. Methodology section should be improved, for example, animal age, N=?; line 111-112 incomplete; "Human ASCs were subcultured in _________ supplemented", several many general mistakes throughout the manuscript. Full cell culturing details are missing i.e. incubator, medium etc. Do the study repeated?
4. Fold change indicated in the figure 1b is not clear and also text size in figure 1c is small and blurr, need to improve.
5. Define figure 2M & 2N in captions. Also, indicate significant differences with line and star (*). It would be nice to add accurate p-values for all the groups.
6. In Figure 5, indicated changes should be marked with arrow/ circle. Figures 5F-K are not clear. Its better to improve the quality. Also, add * and lines to indicate significant differences.
7. English grammar editing required throughout the manuscript.
Author Response
- Abstract should be improved and more structured by considering background, aims & objectives, methodology, results and conclusion. Background is written too long, which is not needed. Add few lines more about results.
Abstract was edited to reduce background.
- Authors have presented their study quite well. Even though, i believe a graphical abstract or a figure showing mechanistic view of the present in the main text would be useful to add.
Graphical abstract included.
- Methodology section should be improved, for example, animal age, N=?; line 111-112 incomplete; "Human ASCs were subcultured in _________ supplemented", several many general mistakes throughout the manuscript. Full cell culturing details are missing i.e. incubator, medium etc. Do the study repeated?
Spelling errors and other errors were corrected. Legends in Figures 2, 6, and 7 were edited to
- Fold change indicated in the figure 1b is not clear and also text size in figure 1c is small and blurr, need to improve.
Figure 1 was edited to increase text size.
- Define figure 2M & 2N in captions. Also, indicate significant differences with line and star (*). It would be nice to add accurate p-values for all the groups.
Captions and p-values were included.
- In Figure 5, indicated changes should be marked with arrow/ circle. Figures 5F-K are not clear. Its better to improve the quality. Also, add * and lines to indicate significant differences.
Labels in Figure 5 were edited to increase font size. P values less than 0.05 were considered significant, and this was identified in the materials and methods section. In addition, each figure legend has a description that states, “Letters not shared indicate a significant difference (p<0.05, Tukey).” Using this lettered scheme is an appropriate way to describe statistical significance without the use of specific p-values for each graph. However, including lines and p-values in bar charts that include all 5 groups is dense and hard to read.
- English grammar editing required throughout the manuscript.
The manuscript was edited.